# 3D Tumor Models in Urology

**DOI:** 10.3390/ijms24076232

**Published:** 2023-03-25

**Authors:** Jochen Neuhaus, Anja Rabien, Annabell Reinhold, Lisa Koehler, Mandy Berndt-Paetz

**Affiliations:** 1Department of Urology, Research Laboratory, University Leipzig, D-04103 Leipzig, Germany; annabell.reinhold@medizin.uni-leipzig.de (A.R.);; 2Department of Urology, Charité-Universitätsmedizin Berlin, Campus Charité Mitte, Charitéplatz 1, D-10117 Berlin, Germany; anja.rabien@charite.de (A.R.); lisa.koehler@charite.de (L.K.); 3Institute of Environmental Science and Geography, University of Potsdam, D-14476 Potsdam, Germany

**Keywords:** 3D cell culture technique, bladder cancer, prostate cancer, renal cell carcinoma

## Abstract

Three-dimensional tumor models have become established in both basic and clinical research. As multicellular systems consisting of tumor and tumor-associated cells, they can better represent tumor characteristics than monocellular 2D cultures. In this review, we highlight the potential applications of tumor spheroids and organoids in the field of urology. Further, we illustrate the generation and characteristics of standardized organoids as well as membrane-based 3D in vitro models in bladder cancer research. We discuss the technical aspects and review the initial successes of molecular analyses in the three major urologic tumor entities: urinary bladder carcinoma (BCa), prostate carcinoma (PCa), and renal cell carcinoma (RCC).

## 1. Introduction

Three-dimensional models, spheroids and organoids, are currently used in oncology research, but also in regenerative research. Our search strategy included Pubmed searches for identification of relevant literature, excluding reviews.

In Pubmed we found 277 original publications (reviews excluded) from 1981 to 2023 using the search term: “urology AND (((“neoplasms” OR “tumor” OR “tumour”) AND (“organ culture techniques” OR “organoids” OR “spheroids”)) NOT review)”. A significant increase in publications was obvious after 2015 (https://pubmed.ncbi.nlm.nih.gov/; accessed on 23 January 2023).

Basically, three different approaches can be distinguished:Patient-derived tumoroids used for optimization of individualized therapies.Organoids for the study of molecular mechanisms responsible for carcinogenesis, tumor spread, and progression; and for testing of new therapies,3D cell culture models in acellular organ matrix for the production of grafts, e.g., for urinary bladder augmentation/replacement.


Traditional 3D cell culture models range from monocultures that develop into organ-like structures using specialized matrices and growth factors [1,2] to cocultures in standard culture plates (e.g., cultivation on a fibroblast feeder layer), to transwell cultures [3]. The advantages of these cultures are their availability and easy handling. For instance, transwell cultures allow analyses of secreted factors from one cell type and their effects on a spatially separated second cell type [4,5].

Spheroids are established from single cell types, mostly fast-growing tumor cells, and cultured either in suspension or in extracellular matrix (ECM), e.g., Matrigel^®^ (CORNING, Corning, NY, USA). The cellular structure of these models largely depends on the cell–cell interactions and the influence of the ECM, thus roles of cellular adhesion molecules and ECM components can be analyzed and/or manipulated. A disadvantage is the unhandiness and the difficult microscopic analysis, requiring special, costly equipment (e.g., confocal laser microscopy or light sheet microscopy).

Organoids on the other hand are 3D cell cultures that combine different cell types to mimic the natural histological structure of organs. Due to the complexity of each organoid, the interactions of the different cell types, the influence of the ECM, and the development and function of spatial compartments can be studied. By using embryonic or induced pluripotent stem cells, organoids that mimic different aspects of the target organs can be generated [6].

Regarding the terminology, organoids are often referred to as multicellular spheroids, which is somewhat confusing since spheroids derived from a single cell type are also multicellular entities. We therefore propose to standardize the nomenclature and refer to spheroids exclusively as free-floating or matrix-embedded 3D cultures from a single cell type (e.g., tumor cells). If multiple cell types are used for construction, these 3D cultures should be referred to as organoids. Various techniques have been developed for the technical production of spheroids and organoids.

## 2. From “Hanging Drop” to “Organ on a Chip”

### 2.1. Hanging Drop

The “hanging drop” method was adapted from microbiology to tissue culture by Ros Granville Harrison as early as 1910, but it still relevant today [7]. Numerous studies have used this method, in which cells are cultured in drops of nutrient solution hanging down from a dish lid [8,9,10,11,12,13]. Although the method is suitable for higher sample throughputs due to the use of 96-well plates, the analytical capabilities are limited due to the inverse culture method. To allow easier handling of spheroids, the method has also been combined with ultra-low attachment well plates (see Section 2.4) [14].

### 2.2. Gel Matrices

Cells are placed in a supportive matrix of various ECM components (e.g., Matrigel: collagen IV, laminin, heparan sulphate proteoglycans, entactin/nidogen), where they develop clonally into spheroids or, if different cell types are used, into organoids [15,16,17]. The cultures are easy to handle and can also be examined microscopically without major problems. Another advantage of this method can certainly be seen in the possibility of forming tubular and branched structures [18]. While commonly used matrices such as Matrigel are derived from animal material, animal-free alternatives are now becoming increasingly popular. Besides avoiding the animal material, they have the advantage of being standardized. Moreover, they can often be used with different degrees of stiffness, e.g., using dextrans that are crosslinked together by a linker such as polyethylene glycol. For subculturing, the cells can then be retrieved from the gel using dextranase [19]. Alternatively, matrices based on self-aggregating peptides (e.g., RADA16-I) have also been prepared [20,21,22]. Further, a combination of gel matrices with transwell systems is also possible.

### 2.3. Porous Scaffolds for Mimicking a Tissue-like Microenvironment

In an innovative approach, scaffolds can simulate a structured matrix for cells to expand in 3D. We used Alvetex Strata (AMSBIO, AMS Biotechnology Ltd., Abingdon, UK), a fine-pored membrane for three-dimensional cultivation of the HT-1376 BCa cells (for details see Appendix B Supplemental Methods). The proliferation dynamics of the cells fixed 3, 13, and 19 days after seeding are illustrated in Figure 1. At day 3, cells are also found in deeper layers, representing a migration from the surface into the membrane. The proliferation rate was about 23% (Figure 1a). In the cross-sectional area of the membrane cultured with cells for 13 days, the proliferation rate was 47% (Figure 1b). After 19 days, the number of proliferating cells in relation to the total amount of cells decreased to 26.8% (Figure 1c). Thus, a proliferation maximum could be determined on day 13 after cell seeding. The cells in the inner region of the membrane tended to migrate through the pores individually and did not exhibit strong cross-linking. Whereas, at the apical side of the membrane, the tumor cells were organized in a multi-layer (Figure 1d).

### 2.4. Ultra-Low Attachment (ULA) Cell Culture Plates

This recently developed 3D cell culture system is available in different formats with up to 384 wells. The cell suspension is centrifuged down to the U-shaped bottoms of the wells and spheroids/organoids form without the need for matrices. This system is also suitable for high-throughput drug screening by using microplate readers or confocal analysis systems (e.g., ImageXpress Micro Confocal system (Molecular Devices, San Jose, CA, USA)). Numerous studies used this new technology [24]. Combinations of ULA plates with matrix gels have also been used [25].

Using this method, we developed a urinary bladder organoid model consisting of malignant or non-malignant urothelial cells (BCa cell lines; HBLAK), bladder fibroblasts, and bladder smooth muscle cells (Figure 2, Figure 3 and Figure 4). The method of organoid generation is described in Appendix B (Supplemental Methods). The organoids showed an “inverse” bladder histology with a multilayered urothelium, and a supportive core of urinary bladder fibroblasts and smooth muscle cells. In the non-malignant organoids using HBLAK cells, immunofluorescence depicts the tight junction-associated zonula occludens-1 (ZO-1) protein in the superficial layer of urothelial HBLAK cells. Vinculin, an adhesion molecule, and part of the cell-ECM interactions, is abundantly expressed in the urothelium as well as in the stromal cells of the organoid core (Figure 2).

The complexity of bladder organoids can even be enhanced by including human umbilical vein endothelial cells into the core of the organoid during the initial 4-day phase of core formation (Figure 3). Those cells form small tubular microvessel-mimics, adding an important structural component to the organoids. First results indicate that vascular cells, represented by HUVEC, may significantly enhance the urothelial differentiation of HBLAK in the outer organoid cell layer.

Standardized bladder organoids generated by the ULA method provide an excellent platform for the analysis of urothelium differentiation, the establishment of a urothelial barrier, and the role of stromal cells (fibroblasts and smooth muscle cells) in these processes. The refinement of analytical methods, such as Western blotting and RT-PCR, is required to overcome the challenge of low cell numbers in the individual organoid and the goal to analyze cell type-specific protein and gene expression. Flow cytometry would be another opportunity. Immunohistological methods including 3D-analysis are appropriate to analyze cell–cell interactions. The option to gradually expand this culture system, e.g., by inclusion of immune cells, to investigate basic immunological processes, is another exciting thought.

We also developed urinary bladder tumor organoids using commercially available BCa cell lines. Within these cultures, the basic features of the bladder carcinoma cells are preserved in the organoids (Figure 4). Tumor organoids also allow the evaluation of anti-tumor treatments in complex environments.

### 2.5. Acellular Matrix Cell Cultures

Usually derived from porcine urinary bladders, an acellular matrix is colonized with different cell types to enable in vivo regeneration of the urinary bladder. Recent studies have shown that in vitro vascularization of the acellular matrix by adipose-derived stem cells is a good prerequisite for in vivo colonization with urinary bladder cells [26]. The aim is to provide biomaterial for urinary bladder augmentation or treatment of urethral strictures [27,28]. Another application is the surgical repair of hypospadias, the long-term success of which is hampered by a lack of a natural tissue bed. Acellular matrix grafts may be a way to overcome these difficulties [29].

### 2.6. Bioprinting

In recent years, so-called bioprinters have been developed to enable computer-assisted design and printing of whole organs [30]. These techniques will be used in the field of transplantation medicine as well as in the field of basic research studies of cellular and molecular mechanisms [31,32,33]. The production of cell-loaded polymer matrices has been described as a urological application [34]. The production of grafts for urinary bladder replacement is complex and currently not in sight due to the histology, the necessary vascularization and innervation [35].

### 2.7. Organ(oid)-on-a-Chip

Kim et al. used bioprinting to fabricate a three-dimensional organoid consisting of urinary bladder carcinoma cells (T-24 or 5637 cells) imprinted on a base of vascular endothelial cells (HUVEC) and lung fibroblasts (MRC-5). The construct was cultured on a permeable membrane and stabilized by a gel cross-linked by UV irradiation. After 48 h of incubation, the organoids were placed in a special microfluidic chamber that allowed separate flow basally, medially, and apically. The authors tested cell viability after apical treatment with BGC (Bacillus Calmette-Guérin) and a basal application of monocytes (THP-1 cell line). The authors demonstrated that BCG significantly reduced cell viability and proliferation of T-24 and 5637 cells. Furthermore, they found a modulation of the concentration of inflammatory cytokines and were able to demonstrate the targeted migration of THP-1 cells into the organoids [36].

This technology is particularly interesting for drug development because screening of substances can be greatly accelerated by numerous tests, such as cytotoxicity analyses in vitro. In personalized medicine, cancer-on-a-chip experiments are the great hopefuls, because the tumor response to different therapies can be tested directly in patients’ cell material before the actual therapy [37,38].

Spheroids and organoids have emerged as the most important tool for replicating complex tumor environments and will be described in more detail below.

## 3. Tumor Microspheroids/Organoids in Urology

Organotypic spheroid cultures allow modeling of organ development and function [39,40]. Technical advances in the development of organoid systems have made it possible to culture cell lines, adult primary cells, or stem and progenitor cells in multicellular units that self-organize and differentiate [40,41]. Organoids in urological research can be divided into cell line-based or patient-derived organoid models.

Several immortalized cell lines can form 3D polarized structures when grown in 3D culture conditions. For example, MDCK (Madin–Darby canine kidney) cells embedded in 3D Matrigel form three-dimensional cyst structures and allow studies of cyst formation or kidney structure [42]. In addition, human bladder carcinoma (BCa) cells were cultured under microcavity conditions in a bioreactor. The BCa cells (ATCC 5637 (HTB-9)) formed 3D constructs that expressed specific markers and structures of a differentiated human urothelium [43]. For the prostate, benign prostatic epithelial cells RWPE-1 seeded in Matrigel were shown to undergo acinar morphogenesis and form consistent 3D structures [44]. Cell line-derived organoids are characterized by high availability and standardization. However, cell lines are susceptible to genotypic changes and loss of tumor heterogeneity during immortalization, which is particularly important for the study of drug effects.

The successful application of 3D organoids derived from adult primary tissues in urological cancer research can further improve our understanding of these diseases. Furthermore, because these primary 3D models retain the genetic background of the original tumor, they can be used as preclinical models for therapeutic patient screening ex vivo. Primary models include patient-derived xenografts (PDXs) or organoids (PDOs), induced pluripotent stem cells (iPSCs), and so-called conditionally reprogrammed cells (CRCs) [45,46].

PDXs are generated by transplantation of human tumor fragments into immunosuppressed mice [45,47]. Because PDX models retain the characteristics of the primary patient tumor, including gene expression profiles and drug responses, they are now used extensively as in vivo models in cancer research [47]. In the field of urology, various types of xenografts have been established for prostate, kidney, and bladder cancers. These PDX models are mainly used for biomarker discovery, tumor differentiation studies, and genetic profiling for novel drug development [48,49]. Despite their advantages, PDX tumor models still have some limitations. First, establishing PDXs is expensive and time-consuming (6–24 months), with success rates varying from 10 to 90% depending on tumor origin and disease characteristics [47,50]. Another limitation of PDXs is the rapid loss of human stromal components, which are replaced by a murine microenvironment during transplantation [51]. Due to these limitations, PDXs are not yet suitable models to make clinical decisions.

PDOs are 3D structures of malignant or non-malignant tissue that spontaneously self-organize in vitro and differentiate to some extent to give rise to functional cell types. PDOs from individual patients retain their genetic aberrations and can reproduce the molecular and histological phenotype of the original tumor in vitro. Therefore, PDOs can be used to study cancer biology and efficiently evaluate patient-specific personalized therapies [46,52,53]. PDOs are genetically manipulable, which gives them an advantage over PDX models [54,55]. In the field of urologic oncology, PDOs from prostate, kidney, and urinary bladder have already been established. However, the limitations of this technology are also evident. For example, prostate cancer and kidney cancer organoids are easily overgrown by non-malignant cells, therefore the success rate of organoid generation from these malignancies is only 15–20% [46,53]. Another problem with PDOs is the lack of nerves, blood vessels, and immune cells to replicate an organoid system [46].

Organoids made from embryonal cells or pluripotent stem cells are another approach to personalized medicine. Inducible pluripotent stem cells (iPSC) are capable of developing into all functional cells of the body [56]. iPSCs are derived from somatic cells by transient expression of transcription factors (e.g., Oct-4, Sox-2, Klf-4, c-Myc), inducing epigenetic remodeling [57,58]. In urology, iPSCs were derived from somatic cells of patients with hereditary renal cell carcinoma, leading to organoids consisting of more than eight different cell types [59]. Gene editing of iPSCs in organoids could be a promising approach in urological research [60]. However, the generation of iPSCs from tumor patients is still time consuming and suffers from low success rates.

The so-called conditional reprogramming (CR) has lately developed into a new platform of artificial generation of tumor cells and non-malignant cell types. For CR, primary malignant or non-malignant cells are kept in co-culture with irradiated mouse fibroblasts in the presence of a rho-associated kinase inhibitor. Under these conditions, the primary cells gain stem cell features such as the ability of unlimited proliferation without the need of exogenous transfection [61]. From conditional reprogrammed cells (CRCs), 3D-cell cultures or organoids can be generated, or they can be cultured as xenotransplants [62,63]. Malignant CRCs might even share specific genetic aberrations and tumor properties with the primary tumor [63,64]. So far, CRCs in urological research were mainly generated from prostate and bladder tumors.

### 3.1. Prostate Carcinoma (PCa)-Models

Among the 3D models in the field of urology, the spheroid, and organoid models of PCa form the predominant part. This is certainly due to the importance of PCa in uro-oncology.

A well-established method is clonogenic cell analysis, which is also suitable for high-throughput analysis using multi-well readers and matrices for cell growth. Using this technique, Muniyan et al. analyzed the efficacy of chemotherapeutics in several PCa cell lines and PCa xenografts and could show that sildenafil enhanced docetaxel-induced prostate cancer cell death [65]. Because of the high seeding efficiency, this method could translate into patient tailored chemotherapy.

As in BCa research, PDX models have also become established as an adjunct to cell culture experiments. However, xenotransplantations of PCa cells into non-humanized immunodeficient mice do not allow for investigation of the involvement of the immune system in tumor progression and its role in therapy.

Nevertheless, essential aspects of PCa tumorigenesis could be elucidated using these models. For example, Fu et al. recently demonstrated that the ovarian tumor deubiquitinase 6A (OTUD6A), which was initially identified in ovarian cancer, is also highly expressed in PCa and is associated with poor prognosis. Overexpression of OTUD6A prevents polyubiquitination and thus stabilizes the expression of the androgen receptor and SWI/SNF ATPase subunit Brg1 (SMARCA4) [66]. An established xenograft model of androgen-dependent PCa is the CWR22 xenograft model and was described as early as 1996 [67]. As shown by Long et al., loss of NCOR2 (nuclear receptor corepressor 2) was associated with significantly shorter relapse-free survival in patients undergoing adjuvant androgen deprivation therapy (ADT). Both altered gene transcription and epigenetic changes appear to be responsible for the development of the lethal neuroendocrine subtype of PCa (NEPC) [68].

Lin and coworkers established a biobank of transplantable PDX from PCa needle biopsies back in 2014 to map the molecular and genetic heterogeneity of PCa. They demonstrated that PDX retained histology as well as genomic alterations and global gene expression of the donor tumors [69]. This biobank can be used to study the mechanisms of transdifferentiation under ADT. As recently shown, androgen-dependent PDX can be converted into spontaneously NEPC-forming, androgen-independent PDX by conditional reprogramming (CR). It was shown that the expression of NEPC genes significantly increased while the expression of androgen receptor signaling pathway genes decreased [70]. This example highlights that despite the immunological limitations, such models can make a significant contribution to understanding the tumor biology of PCa.

Is the cellular heterogeneity of PCa epithelial cells also present in PDOs? This question was addressed by a recent study. The authors used RNA sequencing to examine cellular heterogeneity in biopsies, specimens from prostatectomies, and PDOs. They found that epithelial cells accumulated in the tumor with high expression of androgen signaling pathway genes and a tumor-associated, likely carcinogenesis-associated group, of cells. It was also noticeable that ERG-negative cells, in contrast to ERG-positive cells, had expression patterns comparable to normal luminal epithelial cells. This heterogeneity could also be detected in the PDOs, with cell characteristics and their composition matching those of the parent tumor [71]. Similar results were also reported by Beshiri et al. who created a biobank with PDX/PDO from advanced PCa (mCRPC) [72].

A study of the role of miRNA in PCa demonstrated a concordant effect of overexpression of the miR-183 family (miR-182, -96, 183) in benign prostate cells (ATCC RRWP1 cells), PDX, and organoids from freshly harvested prostate epithelial cells. This miRNA family reduced the expression of the zinc transporter and intracellular zinc levels, which led to induction of tumor-associated signaling pathways, such as adhesion, migration, and wound healing, in all three models [73]. This confirms that fundamental properties of PCa are preserved in PDX and derived organoids and that the mechanisms of tumorigenesis can be studied in vitro in these systems. In addition, the possibility of testing new pharmaceuticals in PDOs arises in terms of reduction of animal experiments.

Another study confirms this view by demonstrating that the regulation of PCa plasticity is regulated by the long non-coding RNA H19 (lncRNA H19) [74]. The neuroendocrine subtype (NEPC) of castration-resistant PCa discussed earlier usually occurs after ADT and is lethal. The lncRNA H19 is highly overexpressed in NEPC and leads to clonal expansion of androgen-independent PCa stem cells as well as protection of these cells from ADT through regulation of histone methylation H3K27me3/H3K4me3 [74]. As already known from ovarian carcinoma, silencing of H3K27me3/H3K4me3-associated gene regions leads to a subpopulation of carcinoma cells with embryonic stem cell properties, thus regulation by lncRNA H19 seems to be a fundamental mechanism of carcinogenesis [75].

The use of PDX/PDO or organoids raises the hope of allowing high-throughput analyses. Meanwhile, special 96-well or 384-well cell culture plates optimized for organoid culture are on the market, which can also be used in standard microplate readers. Currently, more methods are being developed to enable high-throughput analysis of PDX and other organoids/spheroids. One approach is fluorescence-based analysis. For example, Choo et al. were able to analyze morphological changes due to treatment with the PARP inhibitor talazoparib in organoids. For this purpose, Matrigel-embedded prostate cancer PDOs were generated from PDX in a 384-well plate format and analyzed using nuclear staining with Hoechst 33,342 in a Cytation5 Cell Imaging Multi-Mode Reader (BioTek, Winooski, VT, USA). In addition, they used cell-based assays (CellTiter-Glo, Promega, Madison, WI, USA) to measure the number of living cells [76].

### 3.2. Urinary Bladder Carcinoma Models

Both cell line-based and patient-derived organoid models have been successfully established for BCa research. For example, to date, more than 70 PDX models for urothelial carcinoma have been described in the literature, with the majority of PDX lines generated from lower urinary tract malignancies. In urothelial carcinoma, the success rate of establishing xenografts from patients with high-grade disease is higher than that of establishing PDXs from renal tumors or PCa [45]. Urologic PDX models are increasingly used for new biomarker discovery and genomic characterization for new drug development [48,49]. For example, Wei et al. profiled BCa PDXs and identified several somatic nonsense mutations associated with cisplatin resistance [48]. In addition, Cai et al. established a cisplatin-resistant PDX model as well as an associated organoid model and characterized their growth dynamics during subcultivation. Analysis of the transcriptome revealed progression of both tumor models to aggressive phenotypes that exhibited increased proliferative and stem cell-like expression profiles [77].

Patient-derived BCa organoids (PDOs) have also been successfully established from resection samples and have already been used for the development of new therapeutic approaches. The sample spectrum ranged from non-muscle invasive to highly muscle invasive urinary bladder tumors [78,79,80]. Using immunohistochemistry and sequence analysis, it was shown that these resulting BCa organoids contained both basal and luminal subtypes. It was also demonstrated that the PDOs had frequent mutations of certain targets such as TP53 and FGFR3 [78,79]. Lee et al. showed that organoids and corresponding orthotopic xenografts can be converted into one other with high efficiency. This suggests that these models are useful for validating drug responses, drug toxicity, and advancing novel treatment strategies [79]. Wei and colleagues found significant differences in the sensitivities of BCa-cells and PDOs to cisplatin, venetoclax (a new Bcl-2 inhibitor), and S63845 (a novel MCL1 inhibitor). They conclude that PDOs more likely reflect the in vivo situation of BCa, affecting the reliability of preclinical test systems [81]. Yu et al. have already gone a step further in their studies and established PDOs from luminal and basal resection specimens of a muscle-invasive BCa. They identified the CAR (chimeric antigen receptor) target antigen mucin 1 (MUC1) in organoids as well as the corresponding original tumor and produced CAR-T cells against MUC1, which showed significant cytotoxic effects in the organoids [80]. In contrast, Kim and colleagues developed so-called “assembloids” by 3D reconstruction of representative BCa organoids (luminal, basal) co-cultured with the corresponding primary tumor-associated fibroblasts and endothelial cells. By genetic manipulation of PDOs, they demonstrated that FOXA1 in BCa cells (induced by stromal bone morphogenetic protein), is a critical regulator of tumor phenotype formation [82].

Epithelial cells can differentiate into sub-specialized epithelial compartments (such as basal or luminal urothelial cells). Pluripotent stem cells, on the other hand, can form all cellular components of an organ, including epithelial, stromal, and endothelial cells. Three-dimensional culture techniques can be used to generate organoids from embryonic cells or pluripotent stem cells. In 2014, two studies reported the generation of urothelium from human embryonic stem cells and human iPSCs, respectively [83,84]. Osborn et al. developed a matrix and cell contact-free in vitro culture system to differentiate human embryonic stem cells (H9 cell line) or human iPSCs first into endoderm and then into uroplakin-positive urothelium by targeted treatment with a urothelial cell-conditioned medium [83]. In contrast, Kang and colleagues developed a differentiation protocol to differentiate human pluripotent stem cells into urothelial cells using chemically defined culture media [84]. These differentiation protocols may be useful to study the normal and pathological development of human urinary bladder urothelium in vitro. Moreover, they could be used in the field of “tissue engineering” to develop tissue grafts for reconstructive urology.

Reprogrammed cells, CRCs, are suitable for studying the basic biology of BCa or developing new diagnostic and therapeutic methods. Recently, Jiang and colleagues successfully established CRCs from urine samples of BCa patients. The overall success rate for establishing CRCs from urine was more than 80%, with a percentage of 85.4% for CRCs from urine of high-grade BCa patients and 75.0% for low-grade BCa. Sequence analysis confirmed that these urinary CRCs retain the genetic characteristics of the original tumors and therefore may be used for noninvasive diagnosis of BCa [85]. In addition, patient-derived CRCs from tumor tissue have already been established and characterized with respect to their potential for personalized drug screening [64].

In addition to primary patient-derived models, various cell line-based organoids or spheroids have been established. Most 3D BCa models from commercial cell lines have been generated with the use of an artificial extracellular matrix. For example, Boxberger et al. showed that RT-112 cells cultured in alginate formed multicellular spheroids with two to three cell layers and numerous cell–cell contacts [86]. In another study, BCa cell lines cultured in Matrigel (RT-4, RT-112, EJ) were shown to form cell–matrix interactions that affect gene expression and may be important for BCa progression [87]. Smith et al. cultured human BCa cells (ATCC 5637 (HTB-9)) in a bioreactor with a rotating vessel preventing cell attachment. The generated 3D constructs expressed specific urothelial differentiation markers (e.g., E-cadherin, cytokeratin-20, uroplakin-I), and allowed studying of interactions between uropathogenic *E. coli* and the human urothelium [43]. Goulet and colleagues used “tissue engineering” to develop a multilayered stroma equivalent consisting of multiple cell layers of an endothelial cell and fibroblast mixed culture and a urothelial cell layer. Compact BCa spheroids from noninvasive or invasive BCa cell lines (RT-4, T-24) were integrated into this multilayer stroma construct. In this model, invasive BCa cells (T-24) were shown to cross the basement membrane and invade the stromal compartment, whereas non-invasive BCa cells (RT-4) were restricted to the urothelial cell layer [88].

### 3.3. 3D Models of Renal Cell Carcinoma (RCC)

In line with the frequency of clear cell tumors, 3D models for renal cell carcinoma have also largely been developed for clear cell subtypes to date, as the availability of human tumors from the clinic plays a significant role in most models. However, in contrast to many established cell culture methods, the growth rate for 3D models using material from primary tumors or metastases is often low, limiting the use of those models. Nevertheless, several authors were able to establish PDX from primary RCC. PDX models have been described for various subtypes, including clear cell, papillary, sarcomatoid, and translocation renal cell carcinomas [89]. In most cases, the pathological and molecular characteristics were maintained, but they need to be controlled during passages, as genomic alterations may occur during cultivation. PDX models can be applied subcutaneously or orthotopically (i.e., in the renal capsule). In the orthotopic model, the use of 300 µm thick tissue slices results in good attachment rates, good vascularization, infiltration of malignant lymphocytes and metastatic potential, and can be monitored by ultrasound. The growth rate in the orthotopic model is good at 80% for metastases, but only 14% for primary tumors [90]. Compared to traditional cell lines, PDXs are more suitable to study antineoplastic therapeutics, resistance mechanisms, and biomarkers that could inform about sensitivity and response to therapeutics; a listing is provided in Tracey et al. [89]. In the future, the analysis of PDX libraries might be more appropriate for personalized medicine than individual PDX models, because their use is very time consuming and expensive [89]. Since immunodeficient mice are not suitable, humanized mouse models including bladder cancer PDXs have been developed for testing of immunotherapies [91]. In future, these models could significantly improve the suitability of PDX in tumor immunology.

In addition to spheroid models, there are now also some approaches for RCC organoids or tumoroids. For example, spheroids of about 100 µm in size could be obtained from established cell lines after two weeks of culture in stem cell medium [92]. In another study, stem cell-like cells were isolated from ACHN and showed higher clonogenic growth, tumorigenicity and invasion, but even less telomerase activity than the cells of origin [93]. Olofsson et al. used A498 spheroids to demonstrate the advantages of a special multiwell chip suitable for microscopic analysis [94]. This allowed analyses of cell cycle and cell arrangement/density to be analyzed by nuclear staining.

Organoids could be prepared from heterogeneous primary cell assemblies from renal tumor patients, i.e., normal and tumor tissues [95,96]. The cells were cultured in stem cell medium. While normal cells grew for even longer, tumor cells mostly entered a quiescent phase from passage 10 [96]. The latter showed tumor-specific markers such as CAIX and a mixed epithelial and mesenchymal phenotype common in renal cell carcinoma [96]. Na et al. demonstrated preservation of clear cell organoid morphology and higher CAIX and vimentin expression in clear cell RCC organoids compared with 2D cultures [97]. Although culture conditions may have an impact, clear cell organoids generally reflect inter- and intratumoral heterogeneity and subclonal populations. It is even recommended to avoid ROCK inhibitors, which otherwise facilitate growth, because of the VHL deficiency typical in clear cell organisms [98]. To obtain even closer to real tumor growth conditions, the density of the matrix can be increased, such as with a patented RAFT 3D system (Lonza) in which tumoroids are cultured in a collagen/laminin matrix for up to 21 days [99]. A very interesting approach is also the cultivation of very small tumor pieces in collagen-based matrix in the so-called “Air-Liquid Interface” (ALI) culture, in which cell types and their characteristics could also be retained [100]. In this process, the cells in the matrix are in contact with the air on one side upwards and with the nutrients of the medium on the other side via a permeable membrane. The ALI method is particularly suitable for the culture of tissue fragments and the cells of the tumor microenvironment and better oxygenates the organoids [101].

Organoids can be successfully cryopreserved and recultured [98,100,102]. This is particularly important for rarer entities such as hereditary papillary renal cell carcinoma, which could be preserved via iPSCs in a biobank [59], or renal tumors of children [103].

Three-dimensional models of RCC are increasingly used in the study of therapeutics and resistance, but also metastasis. For instance, multikinase inhibitors such as SU11274, foretinib, cabozantinib, and levantinib in combination with everolimus had no effect on normal organoids, but clear cell organoids responded to foretinib and SU11274 with growth inhibition and altered signaling pathways [96]. The effect of a WNT inhibitor was also demonstrated in renal tumoroid and PDX models [104]. ALI cultures by Esser et al. responded differently to cabozantinib or the PD-1 inhibitor nivolumab [100], therefore such systems seem very suitable to map individual patient responses. Reustle et al. used patient-derived 3D-ALI cultures of clear cell renal carcinoma (ccRCC) identifying nicotinamide-N-methyltransferase (NNMT) as a potential new therapeutic target [105]. Coculturing (sunitinib-resistant) renal carcinoma cell lines with fibroblasts, endothelial cells, peripheral blood monocytes, and T cells is also a very interesting approach [106]. Using these 3D cocultures, a multidrug therapy was developed. Thereby, a combination of Rapta-C, metformin, erlotinib, and parthenolide act synergistically at low doses so that non-malignant cells are not damaged [106]. In papillary renal tumors (type 1), hyperactivity of the receptor tyrosine kinase MET is typical, but cannot be reproduced by 2D models. Accordingly, these models do not respond to the MET inhibitor capmatinib [107]. Papillary 3D cultures on HGF producing fibroblasts were sensitive to MET mimicking the situation in vivo [107].

To study metastasis to bone, the renal carcinoma cell line 786-O, which was isolated from bone metastases in a mouse model (Bo-786), was cocultured with preosteoblasts in a 3D model in vitro [108]. While Bo-786 inhibited osteoblast differentiation, agents such as cabozantinib could counteract, i.e., they could potentially be used against osteoclastic processes of renal tumor metastases [108]. In another approach, 3D cultures of renal tumor bone metastases were targeted with natural killer (NK-92) cells [109]. The NK-92 cells were equipped with an “Adapter Chimeric Antigen Receptor” (AdCAR) system and could thus be armed with specific antibodies against certain tumor entities [109].

## 4. The Role of the Tumor Microenvironment (TME) and Extracellular Matrix

It is now known that the ECM plays a much larger role in both tumors and 3D models of tumors than previously thought. It is composed of proteins such as collagens and glycoproteins such as fibronectin, laminin, and proteoglycans [110]. The ECM not only provides cells with a defined structure in which to maintain and move, but also serves as a reservoir for growth factors and influences signaling pathways, e.g., via mechanotransduction. The ECM has a major impact on proliferation, survival, morphology, adhesion, and motility of cells. Angiogenesis is also dependent on basement membrane modifications, for example [110]. Decellularized ECM structures retain much of their growth factors and can induce pluripotent stem cells to differentiate, as shown for renal tissue ECM [111].

During tumorigenesis, fibroblasts in the tumor environment develop into tumor-associated fibroblasts (CAFs) by hypoxia, oxidative stress, growth factors, chemokines, or cytokines, which secrete ECM components to form an abnormal ECM. This is considered to be a “breeding ground” for tumor cells [110]. Most epithelial tumors have a more rigid ECM than corresponding non-malignant tissues, e.g., with more lysyl oxidase cross-linking collagens. The glycosaminoglycan hyaluronic acid from the ECM causes electromechanical swelling in the tumorigenic tissue associated with permeable vascular structures and loss of lymphatic drainage, followed by an increase in interstitial pressure and hypoxia. In addition to influencing inflammatory processes and the mobility or localization of immune cells, ECM is involved in the development of chemoresistance. It is not only a physical barrier for drugs, but also interferes with signaling pathways. As a result, there are therapeutic approaches in research that directly or indirectly address the ECM. In invasive processes, ECM proteins such as collagen, fibronectin, and laminin increase cell motility via interaction with integrins, but the ECM is also degraded by matrix metalloproteinases such as MT1-MMP and MMP-2. These fragments of the ECM may in turn contribute to tumor progression [110]. Recent developments are in bioprinting of epithelial tumor models, for which natural as well as synthetic ECM components and different cell types can be used. This allows replication of a structured tissue with vascularization [112].

Interestingly, in co-cultures of prostate carcinoma cells with fibroblasts, the ECM changed, i.e., components of the basement membrane accumulated compared with single cultures, and the invasiveness of tumor cells increased [113], which means that the interaction with stromal cells is substantial. In urothelial carcinoma of the urinary bladder there is a lot of ECM around the invasive tumor cells. During cell invasion, the ECM components tenascin-C, laminin, collagen, and fibronectin reorganize [114,115]. In contrast to normal bladder tissue, MIBC have an altered ECM structure which is used by migratory and invasively growing cells as a guiding structure. Fibrils within the ECM of MIBC are linearized to a higher degree of organization and facilitate invasion [116]. Compared with non-muscle invasive bladder carcinoma (NMIBC), mRNA expression of collagens is increased in MIBC [117,118]. When collagen 1A1 was inhibited, tumor-progressive properties of bladder carcinoma cells such as proliferation, migration, invasion, epithelial-mesenchymal transition, and TGFb signaling decreased [117]. In bladder carcinoma, the ECM is more rigid than in normal tissue and the content of elastic fibers and glycoproteins such as osteopontin is increased [119]. However, although the ECM of clear cell RCC contains more collagen VI, fibronectin, tenascin-C, and fibrinogen than normal tissue, the stiffness tends to be less indicating that the cross-linking is not very strong [120]. Bond et al. suitably constructed a nine-component matrix for ideal culture of clear cell renal cell carcinoma cells. When this matrix was used together with fibrin, the growth of CAFs was also improved. CAFs cannot grow as well in Matrigel because Matrigel rather resembles the microenvironment in normal tissue [120].

Comparison of the molecular composition of Matrigel with a gelatinous matrix extracted from human uterus benign leiomyoma, called Myogel, revealed clear superiority in human cancer 3D-models, such as HSC-3 cells (oral tongue squamous cell carcinoma cell line) [121] and SqCC/Y1 cells (human buccal mucosa squamous cell carcinoma) [122]. The evaluation of Myogel in urological tumors is still open.

Variable stiffness was created e.g., by Merivaraa et al. [123] using nanofibrillated cellulose hydrogels, in which PC-3 prostate carcinoma cells were successfully grown. A minimum stiffness as well as a sufficient pore network and support by RGD peptides was also reported to be essential for prostate cancer cell growth in 3D scaffolds [124]. When the tubular basement membrane of kidney tubules was stiffened, differentiation markers changed and pro-fibrotic growth factors increased [125] suggesting an impact on carcinogenic processes, too. Millet et al. [126] highlighted the role of CAFs inducing matrix stiffening and EMT in a 3D self-engineered bladder model that displayed tumor-promoting effects of CAFs on normal urothelial cells as well.

For the 786-O and HKCSC cell lines, a stem cell medium and laminin-coated plates were described as optimal for 3D growth when comparing different conditions [127], but it is questionable whether this can be transferred to primary material from surgeries. Collagen deposition is also very important in tumor matrix formation and can be induced by hypoxia-inducible factor (HIF)-1 [128]. A strong tissue-specific influence of ECM was demonstrated using mesodermal progenitor cells that differentiated into nephron cells on decellularized ECM of the kidney, apparently by factors bound to proteoglycans [129]. In another study, decellularized ECM of the liver was used to generate a metastatic environment. Wang et al. cultured the renal carcinoma cell line Caki-1 with hepatocytes in a microfluidic system, a “metastasis-on-a-chip” culture, in which they investigated the effect of a therapeutic agent [130]. There are also interesting developments in replicating the metastatic environment in bone metastases. The ECM in bone contains 90% collagen in fibrils and provides the ideal environment for metastatic cells, e.g., from prostate and kidney tumors. In this process, many cytokines, growth factors, and matrix metalloproteinases favor the formation of a metastatic niche where carcinoma cells readily attach or spread because of fibronectin production by osteoblasts [131]. Despite various models such as bone-on-a-chip, it has been difficult to comprehensively construct the environment of bone metastases. Recent developments in 3D bioprinting try to overcome these limitations by combining tumor cells, stroma cells from the stroma, and the ECM. In addition, there are attempts to include immune cells, and to model vascularization, innervation, and differentiation of the tissue. Viscosity or stiffness of the matrix can be mimicked by modification of matrix materials [132]. Such models are promising in more closely approximating the situation in the patient.

However, the usage of standardized matrices is still a major limitation of the current 3D-models. Future studies should put even more effort into the development of matrices closely mimicking the molecular composition and mechanical properties of human (tumor) tissues.

## 5. Summary/Outlook

In the past decade, 3D models have gained much importance in urological research. The available model systems provide a valuable complement to standard cell culture methods. However, their application in high-throughput analyses is currently at the beginning, so they will not yet replace 2D cell cultures in the medium term. Both PDXs and organoids generated from tumor cell lines have their uses. The value of PDXs/PDOs for the optimization of individualized therapy is currently not predictable, since the generation of PDX/PDO from biopsies succeeds only to a small percentage. Moreover, the expansion of organoids to numbers necessary for drug testing takes too much time—time that tumor patients usually do not have. To date, few studies focus on the analysis of cell–cell interactions that would be possible in histologically complex organoids. It is foreseeable that, depending on the availability of automated, complex (multiple labeling) analysis capabilities, 3D models will in future become more relevant in tumor research, drug development/testing, and other cell biology research fields. In addition, future studies should account for the variable molecular composition and mechanical properties of the ECM by comparing the effects of different animal and human derived ECM.

## Figures and Tables

**Figure 1 ijms-24-06232-f001:**
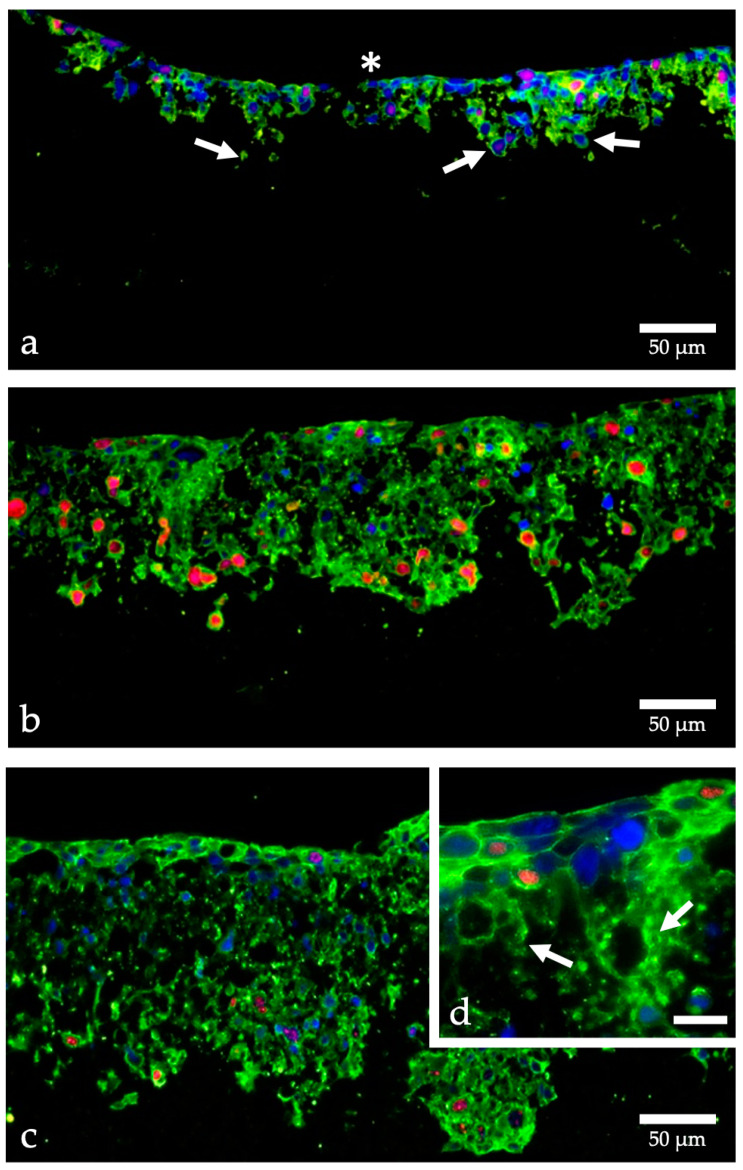
Proliferation of HT-1376 in strata membranes. Cells were grown in Alvetex Strata, fixed with PFA after 3 days (**a**), 13 days (**b**), and 19 days (**c**,**d**), embedded in paraffin, cut into 7 μm thick sections and labeled with fluorescent dyes; cell nuclei blue, beta-actin green, Ki-67 red; (**a**) HT-1376 cells already migrate from the apical side (asterisk) into the membrane (arrows), the proliferation is moderate (red nuclei); (**b**,**c**) many more cells are found within the membrane and the proliferation rate is especially high in those migrating cells; (**d**) higher magnification shows the formation of a superficial multilayered tissue-like structure and indicates the migration of single cells through the membranous pores (arrows); images taken with an CKX53 microscope (Olympus, Hamburg, Germany); calculation of the proliferation rate using Image J. [23]; (**d**) scale bar indicates 20 µm.

**Figure 2 ijms-24-06232-f002:**
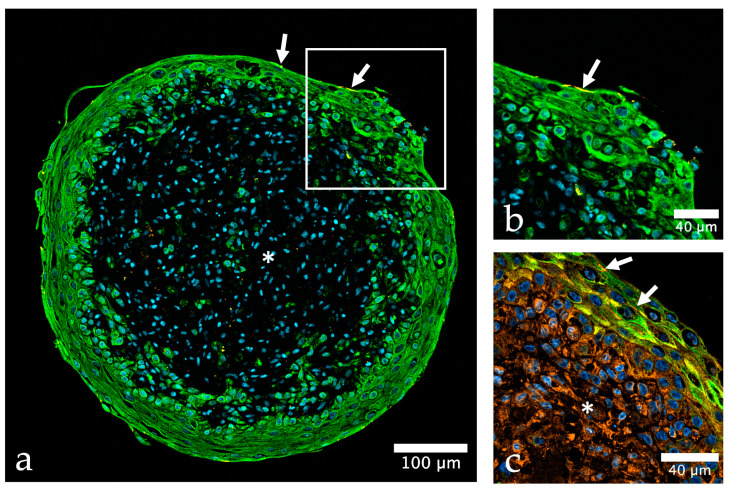
Urinary bladder organoid composed of three different human urinary bladder cell types. (**a**) The organoid shows a clear stratification into outer non-malignant urothelial cells (HBLAK, green) and a core of urinary bladder fibroblasts and smooth muscle cells (HBF, HBSMC); urothelial cells are positive for cytokeratin (green, CKPAN), whereas HBF and HBSMC are negative (asterisk), the tight junction protein zonula occludens-1 (ZO-1) is depicted in orange, nuclei are shown in blue color (stained with TO-PRO-3); (**b**) detail of (**a**) demonstrate that the expression of ZO-1 (orange) is mostly restricted to the top cell layer of HBLAK cells (arrows in (**a**,**b**)), while vinculin (VCL, orange) expression (**c**) is very strong in both, HBLAK and stromal cells (asterisk); antibodies used (Appendix A).

**Figure 3 ijms-24-06232-f003:**
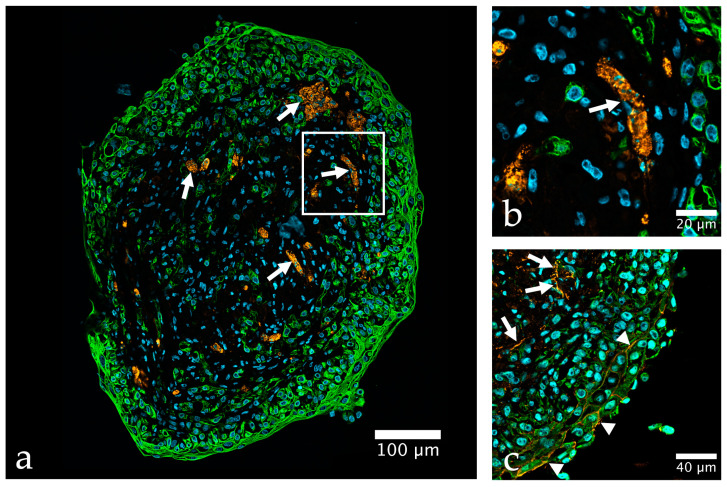
Organoid composed of HBF, HBSMC, HUVEC, and HBLAK, cultured for 14 days with 10 days of differentiation. (**a**) Immunohistochemical staining of cytokeratin 5 & 6 (CK5+6, green), von Willebrand factor (VWF, orange), nuclei are stained with TO-PRO-3 (blue), note the tubular structures formed by HUVEC (arrows); (**b**) detail of (**a**) showing the decoration of HUVEC with VWF; (**c**) zonula occludens protein-1 (ZO-1, orange) indicating barrier formation (arrowheads) by HBLAK stained with an antibody against CKPAN (green), note the labeling of tubular structures within the core of the organoid, probably HUVEC (arrows); antibodies used (Appendix A).

**Figure 4 ijms-24-06232-f004:**
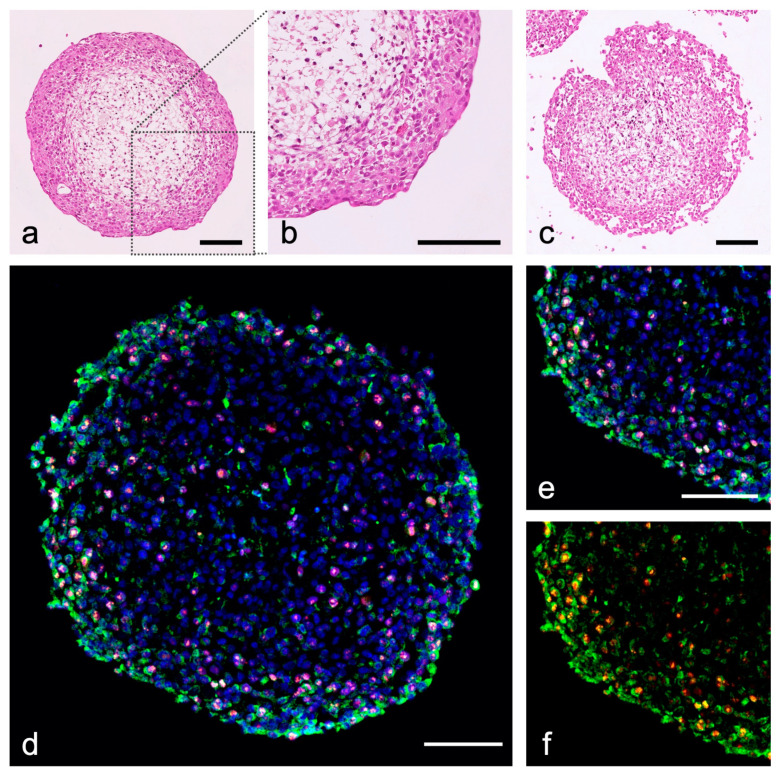
Urinary bladder tumor organoids from RT-112 (initial tumor grade 2) and T-24 (grade 3), HBF and HBSMC. (**a**–**c**) Hematoxylin-eosin staining (HE) of human BCa organoids; (**a**,**b**) RT-112 organoids show a clear urothelial stratification, T-24 organoids show a loose structure (**c**); (**d**–**f**) immunohistochemical staining of cytokeratin (green, CKPAN), Ki-67 (orange), nuclei are stained with DAPI (blue); (**e**,**f**) detail of (**d**) showing T-24 cells by their cytokeratin expression in the organoid (green, CKPAN); T-24 BCa cells show a high proliferation index (red, Ki-67); purple by overlay with nuclei); scale bar: 100 µm; antibodies used (Appendix A).

## Data Availability

Not applicable.

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
