# Peer review of "3D Tumor Models in Urology"

_ijms, 2023, doi:10.3390/ijms24076232_

Round 1
Reviewer 1 Report
The article by Neuhaus et al. describes different established i3D tumor models so far for both basic and clinical research in a very comprehensive manner. As the authors pointed out these 3D culture models will be an asset for oncology research. The review is well through and written well. I have few additional suggestions in the context of ever-changing technology and newer addition.
First aspect the authors need to incorporate is that in vitro 3D model from single cells (PMID: 32847934). These models proved to be a keystone in analyzing the efficacy of cancer treatment agents in 3D environment. This model possesses various characteristics such as 3D construction, slow growth, long follow up period, hypoxic environment and scalable to multicellular architecture I the vicinity of tumor cell derived matrix as opposed to exogenous matric and growth factors and cytokines species non reactivity (PMID: 32847934).
The second aspect the authors need that are incorporate is the ex vivo tumor models, where the fresh tumors are mined into pieces or sliced to study e the therapeutic efficacy or by knocking down interest of gene expression.
Inclusion of 3D culture methods and reagents in details will help the reader. A table of different methods along with reagents, strength and lacunae will be a great addition.
Author Response
Dear Reviewer,
We are grateful for the detailed comments. We tried to thoroughly respond point-by-point. Please also find our text additions in red color in the manuscript.
Sincerely yours,
Jochen Neuhaus
Reviewer 1
The article by Neuhaus et al. describes different established i3D tumor models so far for both basic and clinical research in a very comprehensive manner. As the authors pointed out these 3D culture models will be an asset for oncology research. The review is well through and written well. I have few additional suggestions in the context of ever-changing technology and newer addition.
First aspect the authors need to incorporate is that in vitro 3D model from single cells (PMID: 32847934). These models proved to be a keystone in analyzing the efficacy of cancer treatment agents in 3D environment. This model possesses various characteristics such as 3D construction, slow growth, long follow up period, hypoxic environment and scalable to multicellular architecture I the vicinity of tumor cell derived matrix as opposed to exogenous matric and growth factors and cytokines species non reactivity (PMID: 32847934).
Answer:
Thank you very much for this valuable suggestion. We included this interesting technique in the chapter 3.1 Prostate carcinoma (PCa)-models:
“A well-established method is the clonogenic cell analysis, which is also suitable for high-throughput analysis using multi-well readers and matrices for cell growth. Using this technique, Muniyan et al. analyzed the efficacy of chemotherapeutics in several PCa cell lines and PCa xenografts and could show that sildenafil enhanced the docetaxel-induced prostate cancer cell death {Muniyan et al., 2020, #69637}. Because of the high seeding efficiency, this method could translate into patient tailored chemotherapy.”
The second aspect the authors need that are incorporate is the ex vivo tumor models, where the fresh tumors are mined into pieces or sliced to study e the therapeutic efficacy or by knocking down interest of gene expression.
Answer:
Thank you for this comment. We discussed these models in the chapter “3. Tumor microspheroids / organoids in urology” subsumed as PDO (patient derived organoids). We also described the advantage of genetic manipulation. To further accentuate the methodological advantages of these techniques we added the excellent recent review of Pamarthy and Sabaawy 2021.
“PDOs are genetically manipulable, which gives them an advantage over PDX models {Pamarthy and Sabaawy, 2021, #151089; Hahn et al., 1999, #277777}.”
Inclusion of 3D culture methods and reagents in details will help the reader. A table of different methods along with reagents, strength and lacunae will be a great addition.
Answer: We appreciate your suggestion. However, since we reviewed >100 original papers, usually using more than one technique, we were not able to integrate all the information into one table. Selecting only exemplary papers seemed not justified to us. The methods used in our laboratories are described in the Supplemental Methods. We hope that you can follow our arguments.
Reviewer 2 Report
I appreciate the authors for combining the data as a review. However, here are some comments.
In Section 4 (The potential role of the ECM)
I suggest renaming the section "Role of the tumor microenvironment (TME) and extracellular matrix."
One of the limitations is the studies are not based on microenvironment-mimicking matrices. The authors should mention that as a limitation.
In future direction: please add a section that these studies should be further validated using 3D culture models where the cells (https://doi.org/10.1016/j.yexcr.2018.06.037 ; https://doi.org/10.1186/s12885-015-1944-z ) was analyzed using animal and human-tissue based 3D tissue models. Also, stiffness matters for cell differentiation, which could be considered for future studies.
Author Response
Dear Reviewer,
We are grateful for the detailed comments. We tried to thoroughly respond point by point. Please also find our text additions in red color in the manuscript.
Sincerely yours,
Jochen Neuhaus
Reviewer 2
I appreciate the authors for combining the data as a review. However, here are some comments.
In Section 4 (The potential role of the ECM)
I suggest renaming the section "Role of the tumor microenvironment (TME) and extracellular matrix."
Answer:
Thank you very much for the valuable proposal. We renamed the section accordingly.
4. The role of the tumor microenvironment (TME) and extracellular matrix
One of the limitations is the studies are not based on microenvironment-mimicking matrices. The authors should mention that as a limitation.
Answer:
Thank you for this comment. We added as limitation to:
- The role of the tumor microenvironment (TME) and extracellular matrix
“However, the usage of standardized matrices is still a major limitation of the current 3D-models. Future studies should put even more effort into the development of matrices closely mimicking the molecular composition and mechanical properties of human (tumor) tissues.”
We also added to:
- Summary / Outlook
“In addition, future studies should account for the variable molecular composition and mechanical properties of the ECM by comparing the effects of different animal and human derived ECM.”
In future direction: please add a section that these studies should be further validated using 3D culture models where the cells (https://doi.org/10.1016/j.yexcr.2018.06.037 ; https://doi.org/10.1186/s12885-015-1944-z ) was analyzed using animal and human-tissue based 3D tissue models. Also, stiffness matters for cell differentiation, which could be considered for future studies.
Answer: Thank you very much for this suggestion. We added to:
- The role of the tumor microenvironment (TME) and extracellular matrix
“Comparison of the molecular composition of Matrigel® with a gelatinous matrix extracted from human uterus benign leiomyoma, called Myogel, revealed clear superiority in human cancer 3D-models, such as HSC-3 cells (oral tongue squamous cell carcinoma cell line) {Salo et al., 2015, #212263} and SqCC/Y1 cells (human buccal mucosa squamous cell carcinoma) {Hoque Apu et al., 2018, #119665}. The evaluation of Myogel in urological tumors is still open.
Variable stiffness was created e.g. by Merivaraa et al. {Merivaara et al., 2022, #180100} using nanofibrillated cellulose hydrogels, in which PC-3 prostate carcinoma cells were successfully grown. A minimum stiffness as well as a sufficient pore network and support by RGD peptides was also reported to be essential for prostate cancer cell growth in 3D scaffolds {Bäcker et al., 2016, #254858}. When the tubular basement membrane of kidney tubules was stiffened, differentiation markers changed and pro-fibrotic growth factors increased {Wang et al., 2022, #279217} suggesting an impact on carcinogenic processes, too. Millet et al. {Millet et al., 2022, #269323} highlighted the role of CAFs inducing matrix stiffening and EMT in a 3D self-engineered bladder model that displayed tumor-promoting effects of CAFs on normal urothelial cells as well.”